# Recent Advances in Copper-Based Solid Heterogeneous Catalysts for Azide–Alkyne Cycloaddition Reactions

**DOI:** 10.3390/ijms23042383

**Published:** 2022-02-21

**Authors:** Noura Aflak, Hicham Ben El Ayouchia, Lahoucine Bahsis, Hafid Anane, Miguel Julve, Salah-Eddine Stiriba

**Affiliations:** 1Laboratoire de Chimie Analytique et Moléculaire/*LCAM*, Faculté Polydisciplinaire de Safi, Université Cadi Ayyad, Safi 46030, Morocco; noura.aflak@gmail.com (N.A.); belayou@gmail.com (H.B.E.A.); bahsis.lahoucine@gmail.com (L.B.); ananehafid@gmail.com (H.A.); 2Laboratoire de Chimie de Coordination et d’Analytique/*LCCA*, Département de Chimie, Faculté des Sciences d’El Jadida, Université Chouaïb Doukkali, El Jadida 24000, Morocco; 3Instituto de Ciencia Molecular/*ICMol*, Universidad de Valencia, C/Catedrático José Beltrán 2, 46980 Valencia, Spain; miguel.julve@uv.es

**Keywords:** copper, solid inorganic support, heterogeneous catalyst, click chemistry, cycloaddition reaction, 1,2,3-triazoles

## Abstract

The copper(I)-catalyzed azide−alkyne cycloaddition (CuAAC) reaction is considered to be the most representative ligation process within the context of the “click chemistry” concept. This CuAAC reaction, which yields compounds containing a 1,2,3-triazole core, has become relevant in the construction of biologically complex systems, bioconjugation strategies, and supramolecular and material sciences. Although many CuAAC reactions are performed under homogenous conditions, heterogenous copper-based catalytic systems are gaining exponential interest, relying on the easy removal, recovery, and reusability of catalytically copper species. The present review covers the most recently developed copper-containing heterogenous solid catalytic systems that use solid inorganic/organic hybrid supports, and which have been used in promoting CuAAC reactions. Due to the demand for 1,2,3-triazoles as non-classical bioisosteres and as framework-based drugs, the CuAAC reaction promoted by solid heterogenous catalysts has greatly improved the recovery and removal of copper species, usually by simple filtration. In so doing, the solving of the toxicity issue regarding copper particles in compounds of biological interest has been achieved. This protocol is also expected to produce a practical chemical process for accessing such compounds on an industrial scale.

## 1. Introduction

“Click chemistry” is a concept that uses the most convenient and practical chemical transformations for clicking available reagents or building blocks into final products with high yield and in a selective manner [1]. In 2001, Sharpless listed the criteria of a click chemistry reaction as wide in scope, very high in yield, and stereospecific [1]. Among these “clickable” reactions, the copper-catalyzed [3+2] cycloaddition between azides and alkynes (CuAAC), which is the uncatalyzed version of the known Huisgen reaction usually performed at higher temperatures for longer times, yields a mixture of 1,4- and 1,5-disubstituted 1,2,3-triazoles [2]. Since the independent discovery of CuAAC by Sharpless et al. [3] and Meldal et al. [4], several other research groups have reported an improved version of the Huisgen [3+2] cycloaddition by using a copper(I) catalyst to exclusively produce the corresponding 1,4-disubstituted 1,2,3-triazoles in very high yields with few or no side products. Due to these characteristics, copper-catalyzed azide–alkyne [3+2] cycloaddition (CuAAC) has become a leading method for the covalent assembly of large and small molecules by using 1,2,3-triazole as a bridging block unit [5,6]. Furthermore, transition metals, such as silver [7,8], ruthenium [9], iridium [10], and zinc [11], have also been used to catalyze azide–alkyne cycloaddition reactions for the regioselective synthesis of 1,4- or 1,5-disubstituted or 1,4,5-trisubstituted triazoles (Figure 1).

These five-membered nitrogen heterocyclic scaffolds, especially 1,4-disubstituted 1,2,3-triazoles, are involved in various research areas, including the agrochemical [12], pharmaceutical [13], medicinal [14], and industrial fields [15,16,17,18]. Moreover, 1,4-disubstituted 1,2,3-triazoles have also shown excellent biological properties; for example, as anticancer [19], antiviral [20], and anti-HIV agents [21]. In general, these compounds are prepared through a cycloaddition reaction between alkynes and azides, which depends on reagents (e.g., aliphatic or aromatic) and the selected conditions for a CuAAC reaction, as summarized in Figure 2.

Furthermore, copper(I) species are thermodynamically unstable, and Cu(I) can be easily oxidized to Cu(II) under aerobic conditions. Cu(II) leads to the oxidation of the homocoupling of alkynes in the absence of any reducing agent or ligand [22]. Therefore, the active copper(I) catalysts can be easily obtained by reducing copper(II) precursors with a reducing agent, oxidizing copper metal with copper(II) (comproportionation reaction), or by using a combination of copper salts with a suitable ligand and/or a base [23]. However, the removal of the copper catalyst from the reaction mixture remains a challenge due to its toxicity, which causes serious problems in the synthesis of bioactive molecules [24]. In this regard, great emphasis has been placed on the use of heterogeneous catalysts instead of copper(I)-based homogeneous catalysts in order to facilitate the recovery and recycling of the catalysts, whilst keeping the inherent activity of the catalytic metal center [25].

In recent years, several comprehensive, as well as tutorial, reviews have covered the development of heterogeneous CuAAC methods, mainly based on the immobilization of the copper catalysts as molecular complexes or copper nanoparticles on a variety of supports, including polymers, biopolymers, silica, zeolites, and carbon-based materials [25,26,27,28,29,30,31,32,33,34,35,36,37,38]. The last five-year period has witnessed an increase in the development of such heterogeneous CuAAC methods, taking into account environmental concerns by using water as the reaction medium and avoiding the use of reducing additives, as well as the issue of the recovery of the copper species. The present review provides an up-to-date overview of the covalent and non-covalent immobilization methods of copper salts on inorganic solid supports as well as carbon and magnetic materials applied in azide–alkyne [3+2] cycloaddition reactions, covering the last five years.

## 2. Inorganic Solid-Supported Catalysts for CuAAC

### 2.1. Silica Supports

Silica-supported copper salts play a significant role in heterogeneous catalysis because of their low cost, accessibility, high stability, and large surface area with excellent porosity. They provide the advantage of reaction pathways under mild reaction conditions and comprise environmentally friendly processes toward sustainable chemistry [39]. In this context, Ben El Ayouchia et al. investigated a copper catalyst on silica gel for click chemistry [40]. The Cu(I)-SiO_2_ catalyst was prepared by impregnating dinuclear copper(I) acetate on silica gel in acetone as a solvent. The resulting copper–silica catalyst was found to mediate [3+2] azide–alkyne cycloaddition in 1:1 (v/v) water/ethanol as a mixed solvent, giving rise to a variety of 1,4-disubstituted 1,2,3-triazoles in excellent yields (85–95%) at room temperature. In addition, the immobilized catalyst could be reused up to five times.

The encapsulation of copper by direct interaction with the silica sol-gel matrix loading 9.4 wt.% of copper was also reported by Gil and Coelho et al. in a click reaction [41]. The SiO_2_–Cu composite was prepared using a one-pot sol-gel procedure with the addition of copper iodide during a hydrolysis/condensation reaction of nanofunctionalized tetra-silicon alkylalkoxide. The resulting catalyst was shown to exhibit excellent catalytic activity in the synthesis of 1,4-disubstituted 1,2,3-triazoles under mild conditions using *N*,*N*-diisopropylethylamine (DIPEA) through azide–alkyne cycloaddition reactions in 3:1 (v/v) *tert*-BuOH-H_2_O (90–98% in 3–4 h) or dimethylformamide (DMF) (83–90% in 4–5 h), and through the one-pot three components reactions of halides, sodium azide, and alkynes in 3:1 (v/v) *tert*-BuOH-H_2_O (80–98% in 3–6 h) using 0.05 mol% of catalyst. The recyclability tests showed that this catalyst could be reused for 10 cycles in both click reactions. Hayton et al. [42] reported the synthesis of an organometallic Cu_20_ nanocluster immobilized on silica to catalyze CuAAC reactions. The nanocluster was prepared through the reduction of Cu(OAc)_2_ in the presence of Ph_2_SiH_2_ and phenylacetylene to produce [Cu_20_(CCPh)_12_(OAc)_6_)]. They used 0.5 mol% of catalyst in the click reaction between benzyl azide and phenylacetylene in the CD_2_Cl_2_ as a solvent at room temperature, a process that produced the corresponding compounds in high yield (95%) within 7 h. In a similar approach, Zarenezhad et al. [43] immobilized copper(II) complexes, named [Cu(cdsalMeen)], containing the methyl-2-(1-((2-hydroxybenzylidene)amino)propan-2-yl)amino-1-cyclopentenedithiocarboxylate ligand (cdsalMeen) on silica gel to catalyze the [3+2] cycloaddition reaction between alkynes and *β*-azido alcohols using ascorbic acid as a reducing agent and THF/H_2_O (2:1 v/v) as solvent mixture at room temperature. The *β*-hydroxy-1,2,3-triazolylalkyl products were obtained in good to excellent yields using 0.05 mol% of [Cu(cdsalMeen)]–SiO_2_ catalyst (81–94% in 0.5–1 h). In addition, the catalyst was recycled for five further runs without losing its activity (94–80% within 0.5–1.5 h).

Many studies have focused on the immobilization of copper complexes on inorganic silica supports to improve the stability and selectivity of the copper catalyst on silica gel support. Rhee et al. [44] immobilized Cu(I) and Cu(II) species onto functionalized silica gel (Figure 1). Catalysts **1** and **2** were synthesized via the functionalization of reversed-phase 3-aminopropyl functionalized silica gel by a 2-pyridinecarboxaldehyde ligand in dichloromethane as solvent. The complexation reaction of the resulting Schiff base, named iminopropyl-functionalized silica gel (IPSi), with [Cu(CH_3_CN)_4_]PF_6_ and CuSO_4_ produced Cu(I)@IPSi and Cu(II)@IPSi, respectively. The catalytic activity of the Cu(I)@IPSi and Cu(II)@IPSi composites was investigated in the one-pot three-component condensation reaction of benzyl bromides, phenylacetylenes, and sodium azide, which afforded good to excellent yields (76–97 and 74–96%, respectively) of 1,4-triazoles using 2.5 and 5 mol% of catalyst in water at 60 °C, respectively. Interestingly, the catalysts could be reused for seven consecutive runs with almost equivalent performances. In a similar approach, Bakherad et al. [45] prepared a heterogeneous catalyst by anchoring the Cu(I) aminothiophenol (AT) complex on silica SiO_2_-AT-Cu(I) (**3**) for the synthesis of triazole compounds (Figure 3). The use of 0.2 mol% of SiO_2_-AT-Cu(I) (**3**) catalyst was shown to exhibit a high catalytic activity in one-pot click reactions of aryl bromides, sodium azide, and terminal alkynes in water as solvent at 80 °C within 2–4 h (96–98%). The recovery/recycling tests showed that this catalyst could still be active and reused for at least five repeated cycles (96–89%).

More recently, Bikas et al. reported the functionalization of silica gel with the Cu(II)-hydrazide complex, followed by its use for the construction of *β*-hydroxy-1,2,3-triazoles from the multicomponent CuAAC reaction of alkyl or aryl epoxides, sodium azide, and terminal alkynes (Figure 4) [46]. The catalyst showed high catalytic activity in water at different temperatures, but two products resulted at a temperature higher than 50 °C due to their type of epoxide ring-opening reaction. In addition, the nature of the epoxide reagent was found to have an impressive effect on the resulting products.

In 2017, a copper(II) *bis*-triazole complex immobilized on silica nanoparticles was prepared by Moghadam et al. via the immobilization of the copper(II)-3,5-bis(2-benzothiazolyl)pyridine (Cu(II)Br_2_-BTP) complex on nano-silica functionalized with trimethoxysilylpropylchloride (Cu(II)Br_2_-BTP@TMSP-nSiO_2_) (**5**) (Figure 5) [47]. This heterogeneous catalyst was found to exhibit excellent catalytic activity within short times for the synthesis of 1,4-triazoles regioisomers through one-pot three-component reactions of benzyl halides or α-bromoketones, sodium azide, and alkynes, using a solvent mixture of ethanol/water (3:1 v/v) at 85 °C in the presence of sodium ascorbate. Moreover, the Cu(II)Br_2_-BTP@TMSP-nSiO_2_ catalyst could be reused five consecutive times.

Soltani Rad et al. also reported a new thioamide-based ligand designed for the modification of traditional SiO_2_ to prepare the silica-tethered cuprous acetophenone thiosemicarbazone (STCATSC) (**6**) as a nanocatalyst (Figure 2) [48]. The prepared catalyst **6** was used to synthesize the 1,2,3-triazolyl-based metronidazole hybrid analogues from 2-methyl-5-nitro-1-prop-2-ynyl-1*H*-imidazole with diverse *β*-azidoalcohols in a THF–water medium at room temperature, which provided the products in good to excellent yields (76–92%) within 2 h. Moreover, no significant loss of catalyst activity was noted after five sequential runs. Bai et al. anchored a copper acetylacetonate complex onto an hexagonal mesoporous silica (HMS) solid using 2-butoxy-3,4-dihydropyrans (DP) (Figure 2) [49]. The obtained HMS–DP–Cu(II) (**7**) composite exhibited excellent catalytic activity in the click reaction of sodium azide, organic halides, and alkynes in ethanol medium at 80 °C within 8 h (70–99%).

A supported copper catalyst on 3D-silica functionalized by amino groups was also reported by Gil et al. (Figure 2) [50]. The 3D-SiO_2_–APTS–Cu composite (**8**) was prepared via the surface functionalization of a 3D-printing silica support using (3-aminopropyl)trimethoxysilane (APTS). The obtained 3D-SiO_2_–APTS–Cu catalyst showed high activity for the regioselective synthesis of 1-(2-iodobenzyl)-4-phenyl-1,2,3-triazole from a three-component click reaction involving phenylacetylene, sodium azide, and 2-iodobenzyl bromide in a *tert*-BuOH/H_2_O (3:1 v/v) solvent mixture. Further, it was reused for 10 sequential cycles aimed at investigating its recyclability, and the results revealed that no significant loss of catalytic activity occurred.

Recently, heterogeneous immobilized ionic liquids (ILs) have attracted considerable interest due to their many advantages, such as easy recovery and reusability, low cost, and operational simplicity [51,52,53,54]. In this regard, Moghadam et al. reported silica nanoparticles supporting copper(II)-containing ionic liquid (SNIL-Cu(II)) (**9**) as a new catalyst for click reactions between azides and alkynes (Figure 6) [55]. The SNIL–Cu(II) composite was obtained via the functionalization of silica nanoparticles by 3-chloropropyltrimethoxysilane followed by their complexation with Cu(OTf)_2_, which resulted into composite-loaded 0.21 mmol/g of copper, as measured by ICP analysis. The catalytic activity of catalyst **9** showed high efficiency in a one-pot synthesis of mono- and multi-fold 1,4-disubstituted 1,2,3-triazoles via three-component reactions of alkynes, organic halides, and sodium azide at room temperature in aqueous polyethylene glycol. Moreover, the catalyst **9** was reused six consecutive times.

Similarly, Skoda-Földes et al. [56] supported the copper catalyst on an ionic liquid polymer/silica hybrid composite via the deposition of Cu(I) on an inorganic/organic hybrid material consisting of silica and a polymer of 1-methyl-3-(4-vinylbenzyl)imidazolium chloride (Figure 3). The application of the resulting heterogeneous catalyst (**10)** in CuAAC reactions of alkynes and azides showed moderate to excellent yields (26–99%) in a less polar solvent (dichloromethane) at room temperature within 24 h, using 10 mol% of catalyst. Moreover, the catalyst **10** was able to be used in at least three further cycles. Moghadam et al. [57] attached a poly(*N*-heterocyclic carbine copper complex) onto nano-silica ((Cu(II)-NHCs)n@nSiO_2_) (**11**) (Figure 3). The catalytic activity of the prepared catalyst (**11**) was investigated in the synthesis of 1,2,3-triazoles by the reaction of benzyl halide derivatives, sodium azide, and phenylacetylene. The resulting products were obtained in excellent yields (85–96% in 15–30 min), using 0.02 mol% of the catalyst with sodium ascorbate in an H_2_O/EtOH (2:1 v/v) solvent mixture at room temperature. Moreover, the catalyst **11** was reused up to seven runs, with its activity being sustained (96–79%).

More recently, Berlier et al. [58] prepared novel catalysts based on the organic/inorganic hybrid concept (Figure 4). Six different systems (**12**–**17**) were synthesized, including the Cu(II) catalyst in silica functionalized by β-cyclodextrins (*β*-CD) using diamino (DiAm) and triamino (TriAm) alkoxy silyl spacers, and only two organic–inorganic silica-supported *β*-CD-Cu(II) catalysts were applied in the click azide–alkyne cycloaddition at 85 °C in a *tert*-BuOH/H_2_O (1:1 v/v) solvent mixture. These two Si-TriAm-CD-Cu and Si-DiAm-CD-Cu catalysts exhibited high catalytic activity under greener conditions with short reaction times. The reusability investigation of these catalysts showed that they could be reused up to five and three runs, respectively.

Mesoporous silica, either as Mobil Composition of Matter (MCM-41), Santa Barbara Amorphous (SBA-15), or mesoporous silica materials (KIT-5), are solid supports that provide vast surface areas, making immobilized active species more accessible for substrates during catalytic reactions [59,60,61]. Pourhassan et al. [62] functionalized mesoporous SBA-15 silica by thioamide groups to be applied as a recyclable catalyst for the CuAAC reaction (Figure 5). In this context, the channels of SBA-15 were modified with tris(2-aminoethyl)amine (TAEA) groups and then reacted with S_8_ and phenylacetylene to form thioamide groups. Next, the acquired solid was immersed in a solution of copper(II) chloride to produce the final SBA-15/thioamide-Cu(I) catalyst. The use of this so-formed catalyst (**18**) afforded excellent yields (85–95% in 90–250 min, 1 mol% of catalyst) in synthesizing the desired triazoles from sodium azide, phenylacetylene, and alkyl/benzyl halides or alkyl epoxides in water at room temperature. Furthermore, this catalytic system was able to be used for nine consecutive runs. Banan et al. [63] supported a Cu(II)/Schiff-base complex on SBA-15 to catalyze a one-pot azide–alkyne cycloaddition reaction (Figure 5). The organic–inorganic material was prepared via the immobilization of the complex between 4-[((2-hydroxyethyl)imino)methyl] phenol and CuCl_2_·2H_2_O on a SBA-15 support. The catalyst **19** showed excellent activity in converting organic halides, alkynes, and sodium azide into the corresponding triazole compounds via one-pot azide–alkyne cycloadditions in water within 12 h, and it was also able to be reused for five runs.

Bakherad et al. [64] supported dithizone(dtz)-copper(I) on a SBA-15 support (Figure 7). The obtained SBA-15-dtz-Cu(I) (**20**) material was investigated as a new catalyst in the click reaction between benzyl chloride derivatives, terminal alkynes, and sodium azide in water at 60 °C, affording the corresponding products in good to excellent yields (78–98% in 1–3 h) by using 0.2 mol% of catalyst. In addition, the catalyst **20** was able to be reused over five recycling cycles with yields from 98% to 89% [64]. Sun et al. anchored Cu(OAc)_2_ on a SBA-15 organic/inorganic support via the proton exchange of a carboxyl functionalized SBA-15 silica with copper(II) acetate (Figure 8) [65]. The resulting Cu@SBA-15-PTAA catalyst (**21**) exhibited high catalytic activity for azide–alkyne cycloaddition reactions in water at 50 °C (91–98% in 6–12 h), and it was also able to be reused at least five times.

Rostamnia et al. supported a Cu(I) catalyst on SBA-15 that additionally contained an ethylene diamine (PrEn) and ionic liquid (ImPF_6_) functionalities (Figure 9) [66]. The CuI@SBA-15/PrEn/ImPF_6_ catalyst (**22**) was investigated in the synthesis of 1,4-disubstituted 1,2,3-triazoles from phenylacetylene, sodium azide, and the in situ preparation of azides using two different methods [iodobenzene (method A) and phenylbronic acid (method B)], resulting in excellent yields. Furthermore, the catalyst was able to be recycled for several consecutive runs (7–14). Pzsyan et al. [67] anchored the Cu-Koc acid complex (KA) on functionalized silica-MCM-41 by 3-chloropropyltrimethoxysilane (CPTMS) (MCM41-CPTMS-Kojic acid-Cu) (**23**) (Figure 9). The copper(II) complex on Kojic acid-functionalized nano silica-MCM-41 (**23**) was investigated in the synthesis of 1,2,3-triazoles through the click reaction of 2-(azidomethyl)-5-benzyloxy-4-pyrone and azido Kojic acid with terminal alkynes in ethanol/water (1:1 v/v) and ethylene glycol (EG) as solvents at 100 °C within 5 to 30 min. In addition, this catalyst was able to be recycled up to six times.

Naeimi et al. [68,69] reported a nanocomposite acetylide for click reaction using copper-imprinted periodic mesoporous organosilica (Cu@PMO NC) (Figure 6). This Cu@PMO NC catalyst (**24**) showed excellent results up to 99% for the microwave-assisted synthesis of *β*-hydroxy-1,2,3-triazoles at room temperature in water within 7 min, and was also able to be reused up to six other cycles. M. Heravi et al. [70] supported copper(I) iodide catalyst on modified mesoporous KIT-5 silica via the coordination of 3-aminopropyltriethoxysilane (APTES) on KIT-5 with copper(I) (Figure 6). This catalyst (**25**) exhibited an excellent catalytic performance for the regioselective synthesis of 1,4-disubstituted 1,2,3-triazoles via a one-pot reaction between terminal alkynes, α-halo ketones or alkyl halide, and sodium azide in boiling water (69–88% in 15–150 min). Moreover, the catalyst **25** was reused for six further runs.

### 2.2. Anionic Clay-Supported Catalysts for CuAAC

Hydrocalcites (HTs) are other inorganic heterogeneous catalytic systems that have been described for azide–alkyne click chemistry. Layered double hydroxides or hydrotalcites are one class of anionic clay containing di- M(II) and trivalent M(III) cations, represented by the general formula [M(II)_1−x_M(III)_x_(OH)_2_] (A^n−^)_x/n_.yH_2_O, where A^n−^ is an exchangeable anion in the interlayer usually with water (Figure 7). Layered double hydroxides (LDH) are reported to have attractive properties, such as a simple work-up synthesis, surface hydroxyl groups, a distribution of different metal cations in the brucite layer, intercalated anions with interlayer spaces, biocompatibles, and high chemical and thermal stability. All these properties make them excellent candidates for metal immobilization and catalytic activity [71].

Fülöp et al. [72] reported the Cu(II)Fe(III)-layered double hydroxide catalyst for [3+2] azide–alkyne cycloaddition reactions as a heterogeneous catalyst in a continuous-flow reactor. The catalytic activity of the as-prepared material that was generated from in situ reductions of copper(II) to copper(I) by the oxidative homocoupling of the alkyne (Glaser reaction) exhibited good to high activity (77–98%) in the synthesis of 1,4-disubstituted 1,2,3-triazoles at high pressure/temperature conditions. In 2016, González-Olvera et al. [73] investigated the as-synthesized and calcined (mixed oxide) Cu-Al LDHs. Both catalysts exhibited excellent yields in azide–alkyne cycloadditions in EtOH–H_2_O solvent mixtures under microwave heating. The outcome of the reaction resulted from both hetero- and homogeneous catalytic processes capturing copper(II) from the material by sodium ascorbate. In addition, this study demonstrated that the reconstruction of Cu-Al LDH from Cu(Al)O mixed oxides via the so-called memory effect was not possible, despite the use of severe reaction conditions (EtOH/H_2_O, 80 °C, MW, 10 min). More recently, Cu–Al mixed oxide was also employed for the synthesis of new 1,2,3-triazole derivatives, such as gluco- and allofuranose-linked 1,2,3-triazole derivatives, using glucofuranose and allofuranose diacetonides in the same reaction conditions (EtOH/H_2_O, 80 °C, MW), affording one regioisomer as a product in moderate to good yields (53–89%) within 10 min [74].

Recently, Amini et al. [75] synthesized an LDH-supported copper(0) catalyst via an exchange of [CuCl_4_]^2−^ followed by a reduction with hydrazine hydrate. On exposure to air, LDH-Cu(0) is again oxidized to the LDH-Cu(II) form as a green powder. The prepared LDH-[CuCl_4_]^2−^ catalyst was used in water at 70 °C for the click reaction of terminal alkynes with benzyl azides generated in situ from sodium azide and benzyl halides. Under these conditions, the desired products were obtained in excellent yields (62–96%), and the catalyst was reused for four cycles with relatively high catalytic activity. Bakherad et al. [76] immobilized CuI on Mg-Al-LDH to produce the CuI/Mg-Al-LDH material as a heterogeneous catalyst for CuAAC reactions. Its catalytic activity was investigated in the synthesis of pyrimidine-1,2,3-triazole derivatives in an ethanol medium at room temperature, the desired products being obtained in excellent yields (88–97%) within short reaction times (15–25 min).

### 2.3. Other Inorganic Solid Supports

As shown above, the development of heterogeneous copper catalysts immobilized by solid inorganic supports has attracted attention due to its merger advantages. In the last few years, different approaches have been established to immobilize copper salts onto various solid supports effectively. Table 1 summarizes the catalytic performances of different heterogeneous inorganic solid-supported copper salts in [3+2] cycloaddition reactions by copper salts immobilized onto new alternative inorganic supports.

## 3. Carbon Material-Supported Catalysts for CuAAC

Carbon-based materials are characterized as having good stability, high electrical and thermal conductivities, low production cost, oxidation stability, and low density [85,86,87,88]. In addition to the generally required simple and highly efficient separation issue, these beneficial properties led to their use as support materials in heterogeneous CuAAC. Recently, our group reported the immobilization of copper(I) iodide on activated carbon materials from vegetable biomass through Argan nut shells (Cu-CANS), and the commercially available activated carbon material (Cu-C) was also used for comparative purposes [89]. The Cu-CANS and Cu-C materials were prepared by impregnating the corresponding carbon materials with copper(I) iodide in acetonitrile. The presence of CuI on the carbon surface of the resulting materials was revealed using XRD patterns. The determination of copper loading in the prepared Cu-carbon materials was investigated by using AAS analysis. In this way, 3.82 (Cu-CC) and 1.38 wt% of Cu (Cu-CANS) were found. The Cu-CANS and Cu-C catalysts exhibited moderate to excellent yields (60–95%) in synthesizing the corresponding 1,4-disubstituted 1,2,3-triazoles via [2+3] azide–alkyne cycloaddition in water at room temperature within 6 h using 0.5 mol% of catalyst. Furthermore, these Cu-carbon catalysts were reused ten successive times without any significant loss of activity.

Recently, Devi et al. prepared a SO_3_Cu-carbon catalyst via the immobilization of CuCl_2_ on a glycerol-based SO_3_H-carbon support. Mixed-valence Cu(I)/Cu(II) occurs in this catalyst with a copper loading of 3.32% being determined by X-ray photoelectron spectroscopy (XPS) and industrial plasma optical emission (ICP-OES) analysis, respectively [90]. The obtained catalyst was used in a one-pot CuAAC reaction for the synthesis of β-hydroxy-1,2,3-triazoles in excellent yields (89–94% in 2.5–4 h) using epoxide, sodium azide, and terminal alkyne in water at 60 °C. Moreover, this catalyst was recycled for at least five cycles.

The use of carbon quantum dots (CQDs) as photoinitiators was also investigated in click chemistry through zigzag-structured Cu(I)-doped CQDs for [3+2] cycloaddition between azides and alkynes by Zou et al. [91]. This Cu(I)-CQDs catalyst showed excellent activity in an H_2_O/EtOH (1:1 v/v) solvent mixture under 11 h of irradiation with 365 nm of UV light in the absence of any reducing agent. Furthermore, the obtained results pointed to the fact that the prepared CQDs catalyst support may provide a new opportunity permitting copper(I) to exist in the stable Cu(I)-CQDs catalyst, without decreasing the catalytic activity.

Graphene oxide (GO), a single two-dimensional (2D) layer of carbon atoms, has also been extensively tested in the heterogeneous catalysis of transition metals due to its high surface area and efficient adsorption capacity [92,93,94,95,96]. However, only a limited number of reports are known because of the metal-leaching problems resulting from its weak chelation capacity toward metals. To overcome these problems, research attempts were carried out on GO functionalization. Two strategies were used: (i) the covalent functionalization of the intercalant via coupling on the terminal acid moieties or the opening of epoxides or chemical conjugation on the secondary OH moieties [97,98,99,100]; and (ii) noncovalent interactions on the surface of GO via van der Waals interactions using aromatic substrates [101,102,103,104]. Pourjavadi et al. [105] reported a cross-linked poly(vinyl imidazole) matrix (Pim) on GO as a support for copper(II) salts by complexation with the imidazole rings in the chains. The loading amount of copper ions in the resulting GO/Pim/Cu material was 2.10 mmol/g, calculated by atomic absorption spectroscopy (AAS). This catalyst was used for the regioselective synthesis of 1,2,3-triazole derivatives through a one-pot cycloaddition of halides, terminal alkynes, and sodium azide in a water medium. Good to excellent yields of the products (80–96% in 0.5–3.5 h) were obtained using 1 mol% of catalyst at 50 °C in the presence of sodium ascorbate as a reducing agent. Further, the GO/Pim/Cu catalyst was recovered in up to eight runs without any significant loss of activity. Naeimi et al. functionalized GO via different copper complexes for click chemistry (Figure 10) [106,107,108,109,110]. In 2016, GO was functionalized by 1,7-heptandiamine and isotonic anhydride through a reaction of carboxylic groups on the GO surface with amino groups, and the obtained catalyst (**26**) was treated with CuI (Figure 10) [106,107,108]. Catalyst **26** was investigated in a one-pot cycloaddition reaction of alkyl halides and sodium azide with terminal alkynes in H_2_O/EtOH (1:1 v/v) under different environmental conditions. The desired products were achieved in all cases with good to excellent yields. Furthermore, the recyclability of the catalyst **26** was tested and the results indicate that **it** can be reused several times without any significant deactivation. In 2017, the same group reported a preparation of GO-supported copper catalyst, containing poly-1,2,3-triazoles (PTA) as Cu(I)-stabilizing ligands for the construction of *β*-hydroxy-1,2,3-triazoles (Figure 10) [109,110]. The prepared GO@PTA-Cu catalyst (**27**) was investigated in one-pot three-component reactions between epoxides, terminal alkynes, and sodium azide in water using 0.017 mol% of the catalyst under microwave and thermal conditions, affording the triazole products in excellent yields. Moreover, the catalyst **27** was able to be reused five times without any loss of its catalytic activity.

Recently, a new covalently cross-linked graphene oxide material (GO–CuPPh) was prepared via a cross-linking process using carboxyl groups at the surface of GO with the nucleophilic reaction of copper(II) coordinated by 5,10,15,20-tetrakis(aminophenyl)porphyrin (CuPPh) (Khojastehnezhad et al. [111] (**28**), Figure 11). FT-IR analysis successfully confirmed the covalent linkage of the porphyrin on GO. The amount of copper in the final catalyst was studied using the inductively coupled plasma (ICP) technique, ensuring the successful insertion of copper into the PPh ring with 6.012 wt% of Cu. The catalyst **28** was then tested for the synthesis of 1,4-disubstituted 1,2,3-triazole derivatives via the reaction of aryl azides and terminal alkynes in a H_2_O/EtOH solvent mixture under ultrasonic irradiation at 60 °C with good to excellent yield (88–96% in 5–30 min). In addition, the GO–CuPPh catalyst was able to be reused five times. In a similar approach, Dabiri et al. [112] performed the synthesis of a supported *N*-heterocyclic carbene–copper complex covalently anchored onto graphene oxide (NHC-Cu/GO-IL) through the silylation of GO with the ionic liquid (IL) and, subsequently, the formation of NHC-Cu across the reaction of GO-IL with copper iodide in the presence of *tert*-BuONa as a base. The as-prepared NHC-Cu/GO-IL composite showed excellent results for the synthesis of triazoles (89–99% yields) via reacting terminal alkynes and aryl azides in a H_2_O/EtOH (1:1 v/v) solvent mixture at 70 °C, using 1 mol% of catalyst, within one hour. In addition, no significant reduction in the catalytic activity of this catalyst was observed after ten successive reaction runs. Similarly, Binder et al. [113] immobilized *N*-heterocyclic carbene (NHC)–Cu(I) complexes, prepared from an imidazolium(Ima)-based carbine, on a graphene nanoconjugate (CRGO-Ima-Cu(I)) (**29**) (Figure 8). The catalyst **29** showed excellent catalytic activity in CuAAC reactions (85–99% yields) at 2 mol% catalyst loading within 72 h at 40 °C. It was also able to be reused for up to 10 cycles of click reactions without its decomposition.

In 2018, Ma’mani et al. [114] reported the *β*-cyclodextrin functionalized polyethyleneglycol (PEG)ylated mesoporous silica nanoparticle-graphene oxide hybrid as a support for copper salt (Cu@*β*CD-PEG-mesoGO). The as-prepared nanoparticles were then evaluated for a three-component click reaction of alkynes, benzyl bromides, and sodium azide in water at room temperature. Excellent yields for the synthesis of the corresponding 1,4-triazoles (82–91% in 1–2 h) using 5 mol% of catalyst were achieved. Moreover, Cu@*β*CD-PEG-mesoGO was recovered and recycled up to ten times without any significant decrease in its activity. More recently, Bhagavathsingh et al. [115] reported on the bis(2,2′-bipyridine)copper(II) complex supported on GO (GO@Cu(II)-(bpy)_2_) for the synthesis of 1,4-disubstituted 1,2,3-triazole derivatives. This catalyst was characterized by an XPS analysis that revealed the presence of copper(I) and copper(II) species in GO@Cu(II)-(bpy)_2_ with 3.4% of copper loading, as detected by inductively coupled plasma optical emission spectroscopy (ICP-OES). Next, the nanocomposites were employed for CuAAC reactions using the substrates of azides and acetylene in *tert*-ButOH/H_2_O (2:3 v/v) at room temperature using sodium ascorbate as a reducing agent. The five click reactions showed excellent yields (80–91%) within 60–80 min.

Reduced graphene oxide (r-GO) has also been used to support copper salts in the construction of click products. Navid Soltani Rad et al. reported the Cu/aminoclay (AC)/reduced graphene oxide nanohybrid (Cu/AC/r-GO nanohybrid) (**30**) for the synthesis of 1*H*-1,2,3-triazolyl carbocyclic nucleoside derivatives through CuAAC reactions between the synthesized *N*-propargyl nucleobases and azido alcohols in a THF/H_2_O (1:1 v/v) solvent mixture at room temperature using K_2_CO_3_ as a base (Figure 9) [116]. The catalyst **30** exhibited good to excellent results (80–94% yields in 3–12 h), and was also able to be reused for up to five consecutive runs without any significant loss of activity.

Recently, graphitic carbon nitride (g-C_3_N_4_) has also attracted attention as an excellent support catalyst. This is due to abundant nitrogen functionalities on the surface sites that can act as strong Lewis base sites, while the *N*-bonded planar layered configurations are utilized to anchor the desired metal, allowing its direct use as a heterogeneous catalyst in various reactions [117,118,119,120,121,122,123,124]. Therefore, Payra et al. [125] developed the g-C_3_N_4_ supported copper chloride (Cu@g-C_3_N_4_) (**31**) material as a new heterogeneous catalyst for the one-pot [3+2] cycloaddition (Figure 12). The Cu@g-C_3_N_4_ catalyst exhibited excellent activity in the regioselective preparation of 4-aryl-NH-1,2,3-triazole derivatives via 1,3-dipolar cycloaddition reactions of nitroolefins and sodium azide in water at ambient temperature (yield about 90–99%) in a short reaction time (0.5 h). Furthermore, the synthesized copper catalyst was reused for ten consecutive cycles, and only a minimal leaching (0.08%) was observed.

## 4. Magnetic Solid-Supported Catalysts for CuAAC

The use of paramagnetic supports offers the opportunity for catalyst recovery by using a permanent external magnet. Indeed, this technique can be more advantageous than filtration or centrifugation, justifying the great interest in this approach over the last decade [126,127]. In this context, Hajipour et al. [128] used the CuFeO_2_ nanocatalyst with tetra-*n*-butylammonium bromide for a catalyzed azide–alkyne cycloaddition of phenylacetylene with aryl halides using water as a solvent at room temperature. Excellent yields were obtained (45–92%) within 4 to 24 h, and the catalyst was able to be recycled up to the fourth cycle. Saha et al. synthesized CuFe_2_O_4_ nanoparticles by both sonochemical and mechanochemical processes, and demonstrated that the sonochemical process was better for the uniformity of the sizes of the magnetic nanoparticles (MNPs) [129]. This magnetic CuFe_2_O_4_ NPs species was used as a heterogeneous catalyst for three-component click reactions of alkyl halides, epoxides or boronic acids with sodium azide, and alkynes in water at 80 °C to produce 1,4-disubstituted 1,2,3-triazole derivatives in moderate to excellent yields (55–95%) from 2 to 4 h. In addition, the reported catalyst was able to be reused for five runs without any significant yield decrease. More recently, Gupta et al. [130] reported on 1-butyl-4-methylpyridinium tetrafluoroborate-coated CuFe_2_O_4_ over *L*-tyrosine (*L*-Tyr) functionalized titania nanospheres (IL@CuFe_2_O_4_-*L*-Tyr-TiO_2_/TiTCIL) (**32**) to reduce the aggregation of the nanoparticles in the solution and increase their activity (Figure 10). This magnetically separable nanocatalyst was utilized in a one-pot click reaction of organic bromides, sodium azide, and terminal alkynes. The corresponding products were obtained in good yields (52–95% in 4–10 min) in water at 40 °C. Further, the catalyst showed recyclability for up to seven runs. Eisavi et al. coated CoFe_2_O_4_ with the Cu(OH)_2_ catalyst to reduce the self-aggregation [131]. The resulting CoFe_2_O_4_/Cu(OH)_2_ magnetic nanocatalyst was used in a one-pot synthesis of *β*-hydroxy-1,4-disubstituted 1,2,3-triazole derivatives from aryl or alkyl epoxides, sodium azide, and terminal alkynes in water at 60 °C, with all the reactions being carried out efficiently within 3–6.5 h to give the resulting 1,2,3-triazoles in good yields (75–95%). Furthermore, the catalyst was able to be reused seven times.

In a related approach, many investigations explored the coating of nano-ferrites with a functionalized shell to reduce the aggregation of the nanoparticles in the solution and improve the catalyst’s efficiency by increasing the number of functional groups on the surface. Among these, Moghaddam et al. [132] reported the immobilization of glucose (GLU) onto MNPs of Fe_3_O_4_–silica to stabilize copper(II), and studied their catalytic activity under greener conditions for the one-pot [3+2] cycloaddition reaction of alkyl halides, sodium azide, and alkynes using water as solvent (Figure 11). The use of 5 mol% of (MNPs@GLU)Cl (**33**) catalyst produced excellent yields (76–99% in 1–8 h) in the synthesis of various 1,4-triazole derivatives at 50 °C, due to the high dispersity of the catalyst in water by creating hydrogen bonds with the glucose. Furthermore, the catalyst **33** was able to be recycled and reused for up to seven runs. Jafari et al. reported on the copper(II) acetate/2-aminobenthiol complex (Cu(OAc)_2_-ABT) immobilized on nanoparticles of magnetite-silica (Fe_3_O_4_@SiO_2_/ABT-Cu(OAc)_2_) (**34**). It catalyzed the synthesis of 1,2,3-triazoles via a three-component click reaction of terminal alkynes, alkyl or benzyl halides, and sodium azide in a PEG_400_/H_2_O (9:1 v/v) solvent mixture at room temperature (Figure 11) [133]. The related products were obtained in excellent yields (85–95%) within 10 to 30 min. In addition, Fe_3_O_4_@SiO_2_/ABT-Cu(OAc)_2_ (**34**) was able to be reused for five cycles. Pore et al. described a novel magnetic silica functionalized by 3-aminopropyltriethoxysilane (am), then by acetyl pyridine (ACP), followed by copper coating (CuACP-Am-Fe_3_O_4_@SiO_2_) (**35**) as a catalyst for the one-pot three-component cycloaddition reactions of alkyl or aryl halides, sodium azide, and terminal alkynes in ethanol at 80 °C (Figure 11) [134]. The products were synthesized with excellent yields (82–95% in 15–24 min) using sodium ascorbate as a reducing agent, and the catalyst **35** was able to be recycled six times. Khodabakhshi et al. [135] prepared an immobilized 2-(aminomethyl)benzimidazole-copper(II) complex on Fe_3_O_4_@SiO_2_ nanoparticles to catalyze two different click reactions (Figure 13). This Fe_3_O_4_@SiO_2_@AMBI/Cu (**36**) nanocatalyst effectively catalyzed the click reactions of terminal alkynes and sodium azide with aryl iodide or benzyl halide for the formation of 1,4-disubstituted 1,2,3-triazole derivatives in DMSO)/H_2_O at 100 °C with good to high yields. Zirak et al. [136] prepared a copper(II) complex supported on Fe_3_O_4_@SiO_2_ core-shell MNPs by refluxing a dispersion of Fe_3_O_4_@SiO_2_ and 3-aminopropyltriethoxysilane (APTES) in hexane under N_2_, followed by the treatment of Fe_3_O_4_@SiO_2_-APTES MNPs with 2-cyanopyridine using sodium methoxide in methanol under reflux to give Fe_3_O_4_@SiO_2_ MNPs functionalized with picolinimidoamide (Fe_3_O_4_@SiO_2_-PIA). This last species transformed into the desired picolinimidoamide–copper(II) complex anchored on Fe_3_O_4_@SiO_2_ core-shell MNPs (Fe_3_O_4_@SiO_2_-PIA-Cu) (**37**) via complexation with CuCl_2_.2H_2_O (Figure 12). The catalyst showed moderate to high yields (53–97%) for the synthesis of 1,2,3-triazole compounds via a one-pot cycloaddition reaction of organic halides, sodium azide, and various alkynes at 70 °C in water for 12 h. Moreover, the recoverability of the catalyst was also investigated, and it showed good stability and reactivity for four runs. Mogaddam et al. reported a magnetic heterogeneous catalyst synthesized via the immobilization of copper ion onto the triazole-functionalized Fe_3_O_4_ nanoparticles (Figure 12) [137]. The resulting magnetic catalyst (**38**) was used with sodium ascorbate in the synthesis of 1,2,3-triazoles via a one-pot multicomponent reaction of terminal alkynes, alkyl halides, and sodium azides in a H_2_O/*tert*-BuOH (3:1 v/v) solvent mixture at 55 °C. The catalyst was able to catalyze the desired reactions with excellent yields (85–99%) in 2–7 h, and it was able to be reused ten times. In 2016, the same team reported another MNP catalyst functionalized with α-amidotriazole as a ligand for the immobilization of copper ions on MNPs (MNPs@APTA)Cl_2_ (**39**) (Figure 12) [138]. The prepared catalyst was investigated in a one-pot clicking reaction of alkyl halides, sodium azides, and terminal alkynes in water at 55 °C using sodium ascorbate. The results showed that the corresponding 1,4-triazoles were achieved in excellent yields (82–98% in 2–8 h). In addition, the catalyst **39** was able to be reused for ten reaction cycles. In 2017, the novel magnetic heterogeneous copper catalyst MNPs@8-AQ.CuCl_2_ (**40**) was synthesized via the immobilization of copper(II) chloride onto 8-aminoquinoline (8-AQ)-functionalized magnetic MNPs@8-AQ (Figure 12) [139]. This prepared material showed high catalytic performance for the one-pot three-component synthesis of 1,4-disubstituted 1,2,3-triazole derivatives in the presence of sodium ascorbate and micellar medium at room temperature. The reaction proceeded well for various benzyl/alkyl halides and benzyl/alkyl acetylene, and the corresponding products were obtained in high yields (87–98% in 90–150 min). Furthermore, the catalyst was able to be recycled and reused for up to five runs. Zahmatkesh et al. reported another magnetic catalyst, 1,4-dihydroxyanthraquinone-Cu(II) supported on magnetic Fe_3_O_4_@SiO_2_, for the synthesis of 1-aryl-1,2,3-triazoles [140]. The 1,4-triazoles were isolated in excellent yields (85–97%) through a clicking reaction of the corresponding alkyne, aryl boronic acid derivatives, and NaN_3_ using 0.5 mol% of the catalyst in an H_2_O/MeCN (1:1 v/v) solvent mixture at room temperature within 2.5–6 h. In addition, this heterogeneous catalyst was able to be reused for six consecutive reaction cycles.

In a similar approach, Esmaeilpour et al. synthesized a complex of copper(II) supported on superparamagnetic Fe_3_O_4_@SiO_4_ nanoparticles for the synthesis of 1-aryl-1,2,3-triazole derivatives via the clicking reaction of aryl boronic acid derivatives, alkyne, and sodium azide (Figure 13) [141]. The desired triazoles were obtained in excellent yields (86–97% in 60–120 min) in the presence of 2 mol% of catalyst, and in water as a solvent at 60 °C. The catalyst Fe_3_O_4_@SiO_4_/ligand/Cu(II) (**41**) was also able to be reused for subsequent reactions at least eight times. Zohreh et al. [142] reported a *NNN*-pincer-copper complex immobilized onto MNPs as a heterogeneous catalyst for click reactions (Figure 13). This magnetic catalyst was synthesized through functionalized MNPs via the covalent bonding of 2-aminopyridine to cyanuric chloride followed by complexation with CuI. The finely synthesized catalyst MNP@NNN-pincer(Cu) (**42**) was employed in a one-pot click reaction in water, leading to excellent catalytic activity for the corresponding 1,4-triazoles at 80 °C, and also at room temperature. Moreover, the nanocatalyst was able to be recycled at least eight times.

Mohammadi et al. [143] reported two efficient Cu(I)-nanomagnetic catalysts, namely Fe_3_O_4_@SiO_2_@Schiff base-Cu(I) (**43**) and Fe_3_O_4_@thiourea–Cu(I) (**44**), for the synthesis of 1,2,3-triazoles (Figure 14). Both catalysts were tested in a one-pot three-compound reaction between NaN_3_, alkynes, and boronic acids or alkyl halides in a H_2_O/EtOH solvent mixture at room temperature, and they showed excellent catalytic activity and recyclability up to five runs.

Khodaei et al. reported a novel nanocatalyst that was synthesized via the immobilization of the 4′-(4-hydroxyphenyl)-2,2′:6′,2”-terpyridine/CuI complex on MNPs through a surface modification (FMNPs@SiO_2_-TPy-Cu) (**45**) (Figure 14) [144]. The obtained nanocatalyst was employed for three-component syntheses of 1,4-disubstituted 1,2,3-triazoles from organic halides, sodium azide, and terminal alkynes in a H_2_O/EtOH (1:1 v/v) solvent mixture at room temperature. The resulting products were produced in excellent yields (83–97% in 8–16 h), and the catalyst was able to be recycled for at least five runs. Tabakhsh et al. [145] functionalized nano-magnetite via 2,2′-biimidazole-containing Cu(I) and Cu(II) complexes for a catalyzed click reaction. The prepared nanocatalysts were then investigated for a one-pot synthesis of 1,4-disubstituted 1,2,3-triazoles through the click reaction of halides or tosylates, sodium azide, and terminal acetylenes in water as a solvent at room temperature. The obtained products were obtained in moderate to high yields (65–99%) within 0.5 to 3.5 h. Moreover, the catalyst was able to be easily recovered and reused for up to five consecutive runs.

Recently, *N*-heterocyclic carbenes (NHCs) as linking ligands between magnetic silica and a copper catalyst were also investigated. Misztalewska-Turkowiez et al. prepared a series of NHC-copper complexes immobilized on the surface of MNPs for catalyzed azide–alkyne cycloaddition reactions (**46**–**49**) (Figure 15) [146]. Both supported complexes exhibited good activity in click reactions between the in situ-generated azides with terminal alkynes, but the copper(II) complexes exhibited better activity than those of copper(I). In addition, the catalysts were able to be reused three times.

In a similar approach, the use of *β*-cyclodextrin as a ligand in the anchoring of the copper catalyst to the magnetic silica was also investigated. Shafiee et al. [147] supported copper(I) chloride into *β*-cyclodextrin grafted superparamagnetic iron oxide nanoparticles (Cu@*β*-CD@SPIONs]) (**50**) (Figure 15). This catalyst was reported to exhibit excellent catalytic activity (71–87%) for obtaining 1-benzyl-1*H*-1,2,3- triazoldibenzodiazepinone derivatives via one-pot click reactions in a EtOH/H_2_O (1:1 v/v) solvent mixture at room temperature within 24 h, with the advantage of being able to be reused up to five times. Mahdavi et al. [148] also used the catalyst **50** for a one-pot synthesis of 1,2,3-triazolylquinazolinone derivatives from the reaction of 4-(prop-2-yn-1-yloxy)-benzaldehyde, benzyl azide, and 2-aminobenzamide (Figure 15). In this case, the catalyst **50** also exhibited excellent catalytic activity in constructing related products (79–88%) in water as a solvent at room temperature within 24 h. It was also able to be reused several times for at least ten cycles.

More recently, Khoshneviszadeh et al. [149] reported copper(I) iodide supported on functionalized magnetic-silica core-shell [CuI/Fe_3_O_4_@SiO_2_(TMS-EDTA)] (**51**) nanoparticles for a catalyzed azides–alkynes cycloaddition reaction (Figure 16). This engineered nanocatalyst produced the corresponding triazole products in good to excellent yields that were also able to be reused up to five times.

The immobilization of copper salts onto polymer-coated MNPs is one of the best ways to construct heterogeneous magnetic catalysts with high stability, high loading, and high activity. Several polymer-coated MNPs have been designed and used to support the immobilization of copper salts over the last few decades. Valizadeh et al. reported that MNPs functionalized with poly(4-vinylpyridine) immobilized copper(II) salts [150]. This material was used as a recoverable heterogeneous catalyst for a one-pot synthesis of 1,4-triazoles from terminal alkynes, alkyl halides, and sodium azide. An amount of 2 mol% of this catalyst using sodium ascorbate revealed high yields (73–98%) at 55 °C in a H_2_O/*tert*-BuOH (4:1 v/v) solvent mixture. In addition, this catalytic system was able to be separated and reused for at least nine cycles keeping its main activity. Bahrami et al. [151] immobilized copper(II) salts onto a ferromagnetic nanoparticle triazine dendrimer [FMNP@TD-Cu(II)] (**52**) (Figure 16). The nanoparticles of catalyst **52** were tested as heterogeneous catalysts in a one-pot reaction to prepare 1,4-disubstituted 1,2,3-triazole derivatives from terminal alkynes, organic halides, and sodium azide in the presence of Na_2_CO_3_ as a base and a EtOH-H_2_O (1:1 v/v) solvent mixture at an ambient temperature. This catalytic system produced high yields of the corresponding products (90–99%) within a short time (15–40 min), and it was also able to be reused several times up to six cycles keeping its main activity (99–93%). Sardarian et al. [152] immobilized dendrimer-encapsulated copper(II) salts on Fe_3_O_4_@SiO_2_ NPs (Figure 17). This material (**53**) was used for the synthesis of regioselective 1,4-disubstituted 1,2,3-triazole derivatives via a one-pot three-component reaction of alky halides, sodium azide, and terminal alkynes in excellent yields (70–95% in 2–6 h) at room temperature in water as solvent. Furthermore, the catalyst **53** was able to be reused after eight consecutive recycles. More recently, Taheikal-Koshvandi et al. proposed a novel route to produce magnetic silica by synthesizing silicalite-1 via a sol-gel process followed by a Fe_3_O_4_/silicalite-1 nanocomposite, and finally Fe_3_O_4_/silicalite-1/PVA/Cu(I) produced via an in situ process [153]. The obtained Fe_3_O_4_/silicalite-1/PVA/Cu(I) composite was used in a three-component reaction. It showed excellent catalytic activity to yield the corresponding 1,2,3-triazoles from organic halides, sodium azide, and terminal alkynes in boiling water within 10 min (yield: 87–97%). Moreover, the Fe_3_O_4_/silicalite-1/PVA/Cu(I) catalyst was able to be recovered and directly reused for five runs with excellent activity up to 86%.

Magnetite nanoparticles coated by layered double hydroxides (LDH) instead of silica were also reported. Pazoki et al. [154] synthesized a copper(I)-cysteine complex immobilized on the surface of LDH-coated magnetite nanoparticles, Fe_3_O_4_@LDH@cysteine-Cu(I) (**54**), as a novel recoverable nanocatalyst for the synthesis of 1,4-disubstituted 1,2,3-triazoles (Figure 18). This LDH-based heterogeneous catalyst was able to effectively promote a one-pot reaction between organic halide derivatives, alkynes, and choline azide, both as a solvent and a reagent instead of using sodium azide to achieve the corresponding triazoles in excellent yields (80–97%) at 75 °C. The magnetic catalyst **54** was able to be reused for up to five successive runs without any significant loss of high catalytic performance (90–86%).

In recent years, the immobilization of MNPs on carbon materials, such as activated carbon and carbon nanotubes, has attracted great interest because it is a cheap and easily achieved process. In this context, catalysts based on magnetic carbon materials have also been studied as CuAAC catalyst supports. In this respect, Khalili et al. impregnated magnetic copper ferrite on mesoporous graphitic carbon nitride for the synthesis of the corresponding CuFe_2_O_4_/g-C_3_N_4_ hybrids [155]. The magnetic nanocomposites showed excellent performance as heterogeneous catalysts in the one-pot azide–alkyne cycloaddition reaction between terminal alkynes and alkyl halides or epoxides azide precursors in water as a solvent at 80 °C. Moreover, the CuFe_2_O_4_/g-C_3_N_4_ catalyst was able to be recovered and reused several times up to six catalytic cycles, with its main activity being retained (95–91%). Moeinpour et al. [156] prepared graphene quantum dots (GQDs) modified NiFe_2_O_4_ and used them to stabilize copper(II) (Figure 17). The resulting Cu(II)/GQDs/NiFe_2_O_4_ (**55**) nanomagnetic particles were used for the formation of 1,4-triazole derivatives via [3+2] azide–alkyne cycloaddition reactions in water as a solvent at 60 °C, affording high yields (88–98%) of the corresponding products in short reaction times (7–25 min). In addition, this catalyst was able to be reused in up to five consecutive cycles. In a similar approach, Rafiee et al. supported a mixed Cu(I)/Cu(II) catalyst on a magnetic cysteine functionalized graphene oxide [Cu^I/II^/Cys-MGO] and used it for a one-pot synthesis of 1,4-diaryl-1,2,3-triazoles [157]. The desired products were achieved in good to excellent yields (85–95% in 110–300 min) using H_2_O/EtOH (1:1 v/v) as solvent mixture at 60 °C. Furthermore, the Cu^I/II^/Cys-MGO catalyst was able to be reused up to eight times. Shaabani et al. [158] reported a magnetic guanidine acetic acid (GAA)-functionalized multi-wall carbon nanotube (MWCNT) as a heterogeneous catalyst for click reactions (Figure 17). The catalytic activity of Cu/MWCNT-GAA@Fe_3_O_4_ (**56**) was investigated in azide–alkyne cycloaddition reactions in water as solvent at 50 °C. The results showed a wide range of corresponding 1,2,3-triazoles, which were isolated in good to high yields (82–100% in 1–10 h), and the catalyst **56** was able to be reused four times. Gravel et al. used a functionalized carbon nanotube-copper ferrite as a magnetic catalyst in a one-pot condensation reaction of organic halides, sodium azide, and terminal alkynes at room temperature, and in a H_2_O/EtOH (1:1 v/v) solvent mixture [159]. The triazole products were obtained in moderate to excellent yields (57–92%) within 24 h, and the catalyst was able to be reused for four catalytic cycles without any significant loss of its activity (90–87%). Hamzehloueian et al. [160] reported the synthesis of a magnetic hybrid chitosan/graphene oxide composite as well as its complexation by copper(II) salts. This material was used as a heterogeneous catalyst for a one-pot synthesis of 1,2,3-triazoles from primary halides, sodium azide, and terminal alkynes using sodium ascorbate and a H_2_O/EtOH (1:1 v/v) solvent mixture at 50 °C. The resulting products of the reactions were obtained in high yields (91–98% in 0.25–2 h) and the catalyst was able to be recovered and recycled for nine runs without any significant loss of its activity (98–95%).

## 5. Conclusions

CuAAC is generally considered to be the most creative and successful click reaction leading to 1,4-disubstituted 1,2,3-triazoles in very high yields under mild conditions and with a dramatic rate acceleration. This strategy of making triazole linkers (which are considered more than just passive linkers) through carbon–nitrogen bond formation has also led to many interesting applications in drug discovery, bioconjugation applications, polymer chemistry, and synthesis of natural products. The heterogenization of CuAAC is an excellent strategy to take CuAAC into industrial processes. This tutorial review, as well as other recently published reviews [161], has discussed the prolific methods used to achieve such heterogenization of CuAAC by using the support of catalytically active copper(I) or its copper(II) or copper(0) precursors, which herein includes their immobilization on solid supports, including silica and related structures, carbon materials, and magnetic solids. These approaches have led to the production of efficient heterogeneous catalysts for the click reactions of a variety of functionalized triazoles, as summarized in the present review, under very mild conditions and easy recovery. Therefore, the development of new, cheap, and easily made heterogeneous catalytic systems that allow the performance of CuAAC reactions should take into consideration their use in water at room temperature and in the open air. This would make this excellent ligation process an easily accessible method for non-experts, such as biologists, and a more attractive strategy for the chemical industry manufacturing of pharmaceuticals and fine chemicals on a process scale, with respect to environmental regulations.

## Data Availability

Not applicable.

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
