# Peer review of "Recent Advances in Copper-Based Solid Heterogeneous Catalysts for Azide–Alkyne Cycloaddition Reactions"

_ijms, 2022, doi:10.3390/ijms23042383_

Round 1

Reviewer 1 Report

This is a very good overview on the use of range of copper catalysts in the azide-alkyne cycloaddition reaction.

I have two minor comments: few typos need to be corrected; the referenced number 162 is missing.

I strongly recommend the article for publication.

Author Response

Point-by-point responses to the reviewers:

Reviewer 1:

This is a very good overview on the use of range of copper catalysts in the azide-alkyne cycloaddition reaction.

I have two minor comments: few typos need to be corrected; the referenced number 162 is missing.

I strongly recommend the article for publication.

Authors: We thank this reviewer for the positive comment about our submission and for the recommendation of its publication. We would also like to mention that typos found in the first submitted version were accordingly corrected in the revised version.

The referenced number 162 was replaced by the number 161 (see page 27, conclusion section).

Reviewer 2 Report

The authors briefly reviewed the copper-based solid heterogenous catalysts for azide-alkyne cycloaddition reactions. This work contains some useful information for the related readers and could be considered for publication. However, the authors should modify their manuscript before acceptance for publication according to the following comments:

  1. No conclusive remarks can be seen in the abstract. Hence, the abstract should be re-written.
  2. There are several references cited in Table 1. More references should be cited in Table 1.
  3. It would be better if the authors can show some typical figures describing the catalytic activity of some typical catalysts.
  4. In each section, the authors should summarize the conclusive results after analysis on the cited references, rather than briefly listing the results cited in the references.
  5. It would be better if the authors can envision the development of future research on such a topic.
  6. There are a large number of inappropriate English words or expressions in the manuscript. For example (in abstract), Althoug; perfomed; relys; catalytically; developed; bioisostere… The authors should carefully polish the English of the whole manuscript.

Author Response

Point-by-point responses to the reviewers:

Reviewer 2

The authors briefly reviewed the copper-based solid heterogenous catalysts for azide-alkyne cycloaddition reactions. This work contains some useful information for the related readers and could be considered for publication. However, the authors should modify their manuscript before acceptance for publication according to the following comments:

Authors: We thank also this reviewer for appreciating this article review and for the positive and constructive comments that really helped improving the manuscript.

  1. No conclusive remarks can be seen in the abstract. Hence, the abstract should be re-written.

Authors: Conclusive points have been included in the abstract of the revised version.

  1. There are several references cited in Table 1. More references should be cited in Table 1.

Authors: Just a few articles, and so references, have been published over the last five years concerning the topic of copper salts supported on the other heterogeneous solid supports shown in Table 1.

It would be better if the authors can show some typical figures describing the catalytic activity of some typical catalysts.

Authors: The typical schemes that describe the catalytic activity of some typical heterogeneous catalysts in CuAAC reactions have been now added (see Schemes 3, 7, 11, 12, 17, and 18).

  1. In each section, the authors should summarize the conclusive results after analysis on the cited references, rather than briefly listing the results cited in the references.

Authors: This type of analysis is given in each section, when presenting the results.

  1. It would be better if the authors can envision the development of future research on such a topic.

Authors: Following the reviewer’s suggestion, the development of future research in solid heterogeneous catalysis for the CuAAC reaction has been pointed out, as can be seen in the end of the conclusion section.

  1. There are a large number of inappropriate English words or expressions in the manuscript. For example (in abstract), Althoug; perfomed; relys; catalytically; developed; bioisostere… The authors should carefully polish the English of the whole manuscript.

Authors: All inappropriate English spellings mistakes have been thoroughly checked in the present revised version.